# COMMUNICATION IN MULTI-AGENT REINFORCEMENT LEARNING: INTENTION SHARING

**Woojun Kim, Jongeui Park, Youngchul Sung**[*]
School of Electrical Engineering, KAIST
Daejeon, South Korea
{woojun.kim, jongeui.park, ycsung}@kaist.ac.kr

## ABSTRACT

Communication is one of the core components for learning coordinated behavior in multi-agent systems. In this paper, we propose a new communication scheme named Intention Sharing (IS) for multi-agent reinforcement learning in order to enhance the coordination among agents. In the proposed IS scheme, each agent generates an imagined trajectory by modeling the environment dynamics and other agents' actions. The imagined trajectory is a simulated future trajectory of each agent based on the learned model of the environment dynamics and other agents and represents each agent's future action plan. Each agent compresses this imagined trajectory capturing its future action plan to generate its intention message for communication by applying an attention mechanism to learn the relative importance of the components in the imagined trajectory based on the received message from other agents. Numeral results show that the proposed IS scheme significantly outperforms other communication schemes in multi-agent reinforcement learning.

## 1  INTRODUCTION

Reinforcement learning (RL) has achieved remarkable success in various complex control problems such as robotics and games (Gu et al. (2017); Mnih et al. (2013); Silver et al. (2017)). Multi-agent reinforcement learning (MARL) extends RL to multi-agent systems, which model many practical real-world problems such as connected cars and smart cities (Roscia et al. (2013)). There exist several distinct problems in MARL inherent to the nature of multi-agent learning (Gupta et al. (2017); Lowe et al. (2017)). One such problem is how to learn coordinated behavior among multiple agents and various approaches to tackling this problem have been proposed (Jaques et al. (2018); Pesce & Montana (2019); Kim et al. (2020)). One promising approach to learning coordinated behavior is learning communication protocol among multiple agents (Foerster et al. (2016); Sukhbaatar et al. (2016); Jiang & Lu (2018); Das et al. (2019)). The line of recent researches on communication for MARL adopts end-to-end training based on differential communication channel (Foerster et al. (2016); Jiang & Lu (2018); Das et al. (2019)). That is, a message-generation network is defined at each agent and connected to other agents' policies or critic networks through communication channels. Then, the message-generation network is trained by using the gradient of other agents' policy or critic losses. Typically, the message-generation network is conditioned on the current observation or the hidden state of a recurrent network with observations as input. Thus, the trained message encodes the past and current observation information to minimize other agents' policy or critic loss. It has been shown that due to the capability of sharing observation information, this kind of communication scheme has good performance as compared to communication-free MARL algorithms such as independent learning, which is widely used in MARL, in partially observable environments.

In this paper, we consider the following further question for communication in MARL:

> "How to harness the benefit of communication beyond sharing partial observation."

We propose *intention* of each agent as the content of message to address the above question. Sharing intention using communication has been used in natural multi-agent systems like human society.

---

[*]Corresponding author

For example, drivers use signal light to inform other drivers of their intentions. A car driver may slow down if a driver in his or her left lane turns the right signal light on. In this case, the signal light encodes the driver's intention, which indicates the driver's *future* behavior, not current or past observation such as the field view. By sharing intention using signal light, drivers coordinate their drive with each other. In this paper, we formalize and propose a new communication scheme for MARL named Intention sharing (IS) in order to go beyond existing observation-sharing schemes for communication in MARL. The proposed IS scheme allows each agent to share its intention with other agents in the form of encoded imagined trajectory. That is, each agent generates an imagined trajectory by modeling the environment dynamics and other agents' actions. Then, each agent learns the relative importance of the components in the imagined trajectory based on the received messages from other agents by using an attention model. The output of the attention model is an encoded imagined trajectory capturing the intention of the agent and used as the communication message. We evaluate the proposed IS scheme in several multi-agent environments requiring coordination among agents. Numerical result shows that the proposed IS scheme significantly outperforms other existing communication schemes for MARL including the state-of-the-art algorithms such as ATOC and TarMAC.

## 2  RELATED WORKS

Under the asymmetry in learning resources between the training and execution phases, the framework of centralized training and decentralized execution (CTDE), which assumes the availability of all system information in the training phase and distributed policy in the execution phase, has been adopted in most recent MARL researches (Lowe et al. (2017); Foerster et al. (2018); Iqbal & Sha (2018); Kim et al. (2020)). Under the framework of CTDE, learning communication protocol has been considered to enhance performance in the decentralized execution phase for various multi-agent tasks (Foerster et al. (2016); Jiang & Lu (2018); Das et al. (2019)). For this purpose, Foerster et al. (2016) proposed Differentiable Inter-Agent Learning (DIAL). DIAL trains a message-generation network by connecting it to other agents' Q-networks and allowing gradient flow through communication channels in the training phase. Then, in the execution phase the messages are generated and passed to other agents through communication channels. Jiang & Lu (2018) proposed an attentional communication model named ATOC to learn when to communicate and how to combine information received from other agents through communication based on attention mechanism. Das et al. (2019) proposed Targeted Multi-Agent Communication (TarMAC) to learn the message-generation network in order to produce different messages for different agents based on a signature-based attention model. The message-generation networks in the aforementioned algorithms are conditioned on the current observation or a hidden state of LSTM. Under partially observable environments, such messages which encode past and current observations are useful but do not capture any future information. In our approach, we use not only the current information but also *future information* to generate messages and the weight between the current and future information is adaptively learned according to the environment. This yields further performance enhancement, as we will see in Section 5.

In our approach, the encoded imagined trajectory capturing the intention of each agent is used as the communication message in MARL. Imagined trajectory was used in other problems too. Racanière et al. (2017) used imagined trajectory to augment it into the policy and critic for combining model-based and model-free approaches in single-agent RL. It is shown that arbitrary imagined trajectory (rolled-out trajectory by using a random policy or own policy) is useful for single-agent RL in terms of performance and data efficiency. Strouse et al. (2018) introduced information-regularizer to share or hide agent's intention to other agents for a multi-goal MARL setting in which some agents know the goal and other agents do not know the goal. By maximizing (or minimizing) the mutual information between the goal and action, an agent knowing the goal learns to share (or hide) its intention to other agents not knowing the goal in cooperative (or competitive) tasks. They showed that sharing intention is effective in the cooperative case.

In addition to our approach, Theory of Mind (ToM) and Opponent Modeling (OM) use the notion of intention. Rabinowitz et al. (2018) proposed the Theory of Mind network (ToM-net) to predict other agents' behaviors by using meta-learning. Raileanu et al. (2018) proposed Self Other-Modeling (SOM) to infer other agents' goal in an online manner. Both ToM and OM take advantage of predicting other agents' behaviors capturing the intention. One difference between our approach and

the aforementioned two methods is that we use communication to share the intention instead of inference. That is, the agents in our approach allow other agents to know their intention directly through communication, whereas the agents in ToM and OM should figure out other agents' intention by themselves. Furthermore, the messages in our approach include future information by rolling out the policy, whereas ToM and CM predict only the current or just next time-step information.

## 3  SYSTEM MODEL

We consider a partially observable $N$-agent Markov game (Littman (1994)) and assume that communication among agents is available. At time step $t$, Agent $i$ observes its own observation $o_t^i$, which is a part of the global environment state $s_t$, and selects action $a_t^i \in \mathcal{A}_i$ and message $m_t^i \in \mathcal{M}_i$ based on its own observation $o_t^i$ and its own previous time step message $m_{t-1}^i$ plus the received messages from other agents, i.e., $m_{t-1} = (m_{t-1}^1, \cdots, m_{t-1}^N)$. We assume that the message $m_t^i$ of Agent $i$ is sent to all other agents and available at other agents at the next time step, i.e., time step $t+1$. The joint actions $a_t = (a_t^1, \cdots, a_t^N)$ yield the next environment state $s_{t+1}$ and rewards $\{r_t^i\}_{i=1}^N$ according to the transition probability $\mathcal{T} : \mathcal{S} \times \mathcal{A} \times \mathcal{S} \to [0, 1]$ and the reward function $R^i : \mathcal{S} \times \mathcal{A} \to \mathbb{R}$, respectively, where $\mathcal{S}$ and $\mathcal{A} = \prod_{i=1}^N \mathcal{A}^i$ are the environment state space and the joint action space, respectively. The goal of Agent $i$ is to find the policy $\pi^i$ that maximizes its discounted return $R_t^i = \sum_{t'=t}^\infty \gamma^{t'} r_{t'}^i$. Hence, the objective function of Agent $i$ is defined as $J_i(\pi^i) = \mathbb{E}_\pi [R_0^i]$, where $\pi = (\pi^1, \cdots, \pi^N)$ and $\gamma \in [0, 1]$ are the joint policy and the discounting factor, respectively.

## 4  THE PROPOSED INTENTION SHARING SCHEME

The key idea behind the IS scheme is that multiple agents communicate with other agents by sending their implicit future plans, which carry their intention. The received messages capturing the intention of other agents enable the agent to coordinate its action with those of other agents. We now describe the architecture of the proposed IS scheme. At time step $t$, Agent $i$ selects an action $a_t^i \sim \pi^i(\cdot|o_t^i, m_{t-1})$ and a message $m_t^i = MGN^i(o_t^i, m_{t-1}, \pi^i)$ based on its own observation $o_t^i$ and received messages $m_{t-1}$, where $MGN^i$ is the message-generation network (MGN) of Agent $i$. The MGN consists of two components: Imagined trajectory generation module (ITGM) and attention module (AM). Each agent generates an imagined trajectory by using ITGM and learns the importance of each imagined step in the imagined trajectory by using AM. The output of AM is an encoded imagined trajectory reflecting the importance of imagined steps and is used as the communication message. The overall architecture of the proposed IS scheme is shown in Fig. 1. In the following we describe the detail of each module.

### 4.1  IMAGINED TRAJECTORY GENERATION MODULE (ITGM)

The role of ITGM is to produce the next imagined step. ITGM takes the received messages, observation, and action as input and yields the predicted next observation and predicted action as output. By stacking ITGMs, we generate an imagined trajectory, as shown in Fig. 1. For Agent $i$ at time step $t$, we define an $H$-length imagined trajectory as

$$\tau^i = (\tau_t^i, \hat{\tau}_{t+1}^i, \cdots, \hat{\tau}_{t+H-1}^i), \tag{1}$$

where $\hat{\tau}_{t+k}^i = (\hat{o}_{t+k}^i, \hat{a}_{t+k}^i)$ is the imagined step at time step $t+k$. Note that $\tau_t^i = (o_t^i, a_t^i)$ is the true values of observation and action, but the imagined steps except $\tau_t^i$ are predicted values.

ITGM consists of a roll-out policy and two predictors: Other agents' action predictor $f_a^i(o_t^i)$ (we will call this predictor simply action predictor) and observation predictor $f_o^i(o_t^i, a_t^i, a_t^{-i})$. First, we model the action predictor which takes the observation as input and produces other agents' predicted actions. The output of the action predictor is given by

$$f_a^i(o_t^i) = (\hat{a}_t^1, \cdots, \hat{a}_t^{i-1}, \hat{a}_t^{i+1}, \cdots, \hat{a}^N) =: \hat{a}_t^{-i} \tag{2}$$

Note that the action predictor can be trained by the previously proposed opponent modeling method (Rabinowitz et al. (2018); Raileanu et al. (2018)) and can take the received messages as input. Next,

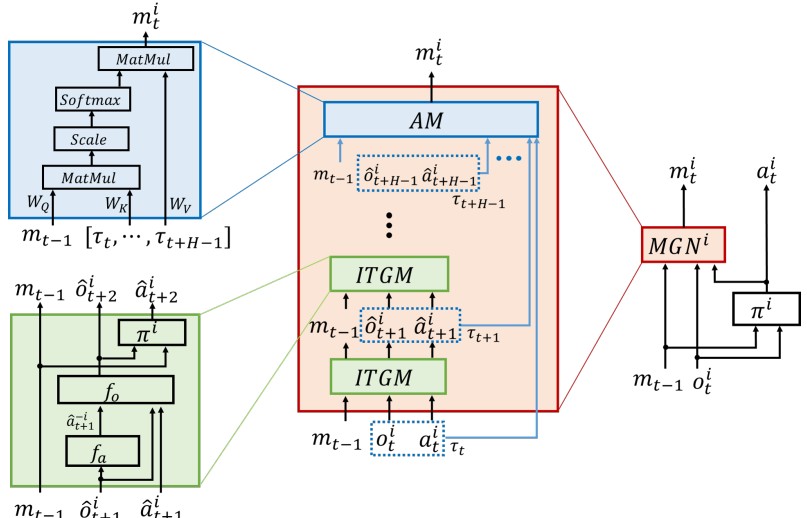

Figure 1: Overall architecture of the IS scheme from the perspective of Agent $i$

we model the observation predictor $f_o^i(o_t^i, a_t^i, \hat{a}_t^{-i})$ which is conditioned on the observation $o_t^i$, own action $a_t^i$, and the output of the action predictor $\hat{a}_t^{-i}$. Here, we adopt the dynamics function that predicts the difference between the next observation and the current observation, i.e., $o_{t+1}^i - o_t^i$ instead of the next observation $o_{t+1}^i$ proposed in (Nagabandi et al. (2018)) in order to reduce model bias in the early stage of learning. Hence, the next observation can be written as

$$\hat{o}_{t+1}^i = o_t^i + f_o^i(o_t^i, a_t^i, \hat{a}_t^{-i}). \tag{3}$$

By injecting the predicted next observation and the received messages into the roll-out policy in ITGM, we obtain the predicted next action $\hat{a}_{t+1}^i = \pi^i(\hat{o}_{t+1}^i, m_{t-1})$. Here, we use the current policy as the roll-out policy. Combining $\hat{o}_{t+1}^i$ and $\hat{a}_{t+1}^i$, we obtain next imagined step at time step $t+1$, $\tau_{t+1}^i = (\hat{o}_{t+1}^i, \hat{a}_{t+1}^i)$. In order to produce an $H$-length imagined trajectory, we inject the output of ITGM and the received messages $m_{t-1}$ into the input of ITGM recursively. Note that we use the received messages at time step $t$, $m_{t-1}$, in every recursion of ITGM.[1]

### 4.2 ATTENTION MODULE (AM)

Instead of the naive approach that uses the imagined trajectory $[\tau_t, \cdots, \tau_{t+H-1}]$ directly as the message, we apply an attention mechanism in order to learn the relative importance of imagined steps and encode the imagined trajectory according to the relative importance. We adopt the scale-dot product attention proposed in (Vaswani et al. (2017)) as our AM. Our AM consists of three components: query, key, and values. The output of AM is the weighted sum of values, where the weight of values is determined by the dot product of the query and the corresponding key. In our model, the query consists of the received messages, and the key and value consist of the imagined trajectory. For Agent $i$ at time step $t$, the query, key and value are defined as

$$q_t^i = W_Q^i m_{t-1} = W_Q^i \left[ m_{t-1}^1 \| m_{t-1}^2 \| \cdots \| m_{t-1}^{N-1} \| m_{t-1}^N \right] \in \mathbb{R}^{d_k} \tag{4}$$

$$k_t^i = \left[ W_K^i \tau_t, \cdots, \underbrace{W_K^i \tau_{t+h-1}}_{=:k_t^{i,h}}, \cdots, W_K^i \tau_{t+H-1} \right] \in \mathbb{R}^{H \times d_k} \tag{5}$$

$$v_t^i = \left[ W_V^i \tau_t, \cdots, \underbrace{W_V^i \tau_{t+h-1}}_{=:v_t^{i,h}}, \cdots, W_V^i \tau_{t+H-1} \right] \in \mathbb{R}^{H \times d_m}, \tag{6}$$

---

[1]Although the fixed received messages cause bias, it is observed that the prediction of received messages generates more critical bias in simulation. Hence, we use $m_{t-1}$ for all $H$ prediction steps.

where $W_Q^i \in \mathbb{R}^{d_k \times N d_m}$, $W_K^i \in \mathbb{R}^{d_k \times d_\tau}$ and $W_V^i \in \mathbb{R}^{d_m \times d_\tau}$ are learnable parameters and operation $\|$ denotes the concatenation of vectors. The output $m_t^i$ of the attention model, which is used for message, is the weighted sum of the values:

$$m_t^i = \sum_{h=1}^{H} \alpha_h^i v_t^{i,h}, \tag{7}$$

where the weight vector $\alpha^i = (\alpha_1^i, \cdots, \alpha_H^i)$ is computed as

$$\alpha^i = \text{softmax}\left[ \frac{q_t^{i^T} k_t^{i,0}}{\sqrt{d_k}}, \cdots, \underbrace{\frac{q_t^{i^T} k_t^{i,h}}{\sqrt{d_k}}}_{=: \alpha_h^i}, \cdots, \frac{q_t^{i^T} k_t^{i,H}}{\sqrt{d_k}} \right]. \tag{8}$$

The weight of each value is computed by the dot product of the corresponding key and query. Since the projections of the imagined trajectory and the received messages are used for key and query, respectively, the weight can be interpreted as the relative importance of imagined step given the received messages. Note that $W_Q$, $W_K$ and $W_V$ are updated through the gradients from the other agents.

## 4.3 Training

We implement the proposed IS scheme on the top of MADDPG (Lowe et al. (2017)), but it can be applied to other MARL algorithms. MADDPG is a well-known MARL algorithm and is briefly explained in Appendix A. In order to handle continuous state-action spaces, the actor, critic, observation predictor, and action predictor are parameterized by deep neural networks. For Agent $i$, let $\theta_\mu^i, \theta_Q^i, \theta_o^i$, and $\theta_a^i$ be the deep neural network parameters of actor, critic, observation predictor, and action predictor, respectively. Let $W^i = (W_Q^i, W_K^i, W_V^i)$ be the trainable parameters in the attention module of Agent $i$. The centralized critic for Agent $i$, $Q^i$, is updated to minimize the following loss:

$$L_Q(\theta_Q^i) = \mathbb{E}_{x,a,r^i,x'}\left[ (y^i - Q^i(x,a))^2 \right], \quad y^i = r^i + \gamma Q^{i^-}(x',a')|_{a^{j'} = \mu^{i^-}(o'^j, m)}, \tag{9}$$

where $Q^{i^-}$ and $\mu^{i^-}$ are the target Q-function and the target policy of Agent $i$ and parameterized by $\theta_\mu^{i^-}, \theta_Q^{i^-}$, respectively. The policy is updated to minimize the policy gradient loss:

$$\nabla_{\theta_\mu^i} J(\theta_\mu^i) = \mathbb{E}_{x,a}\left[ \nabla_{\theta_\mu^i} \mu^i(o^i, m) \nabla_{a^i} Q^i(x,a)|_{a^i = \mu^i(o^i, m)} \right] \tag{10}$$

Since the MGN is connected to the agent's own policy and other agents' policies, the attention module parameters $W^i$ are trained by gradient flow from all agents. The gradient of Agent $i$'s attention module parameters is given by $\nabla_{W^i} J(W^i) =$

$$\frac{1}{N} \sum_{j=1}^{N}\left[ \mathbb{E}_{\overline{x}, \overline{m}, x, a}\left[ \nabla_{W^i} MGN(\tilde{m}^i | \overline{o}^i, \overline{m}) \nabla_{\tilde{m}^i} \mu^j(o^j, \tilde{m}^i, \tilde{m}^{-i}) \nabla_{a^j} Q^j(x,a)|_{a^j = \mu^j(o^j, m)} \right] \right], \tag{11}$$

where $\overline{o}^i$ and $\overline{m}$ are the previous observation and received messages, respectively. The gradient of the attention module parameters are obtained by applying the chain rule to policy gradient.

Both the action predictor and the observation predictor are trained based on supervised learning and the loss functions for agent $i$ are given by

$$L(\theta_a^i) = \mathbb{E}_{o^i, a}\left[ (f_{\theta_a^i}^i(o^i) - a^{-i})^2 \right] \tag{12}$$

$$L(\theta_o^i) = \mathbb{E}_{o^i, a, o'^i}\left[ ((o'^i - o^i) - f_{\theta_o^i}^i(o^i, a^i, \hat{a}^{-i}))^2 \right]. \tag{13}$$

## 5 Experiment

In order to evaluate the proposed algorithm and compare it with other communication schemes fairly, we implemented existing baselines on the top of the same MADDPG used for the proposed scheme.

The considered baselines are as follows. 1) MADDPG (Lowe et al. (2017)): we can assess the gain of introducing communication from this baseline. 2) DIAL (Foerster et al. (2016)): we modified DIAL, which is based on Q-learning, to our setting by connecting the message-generation network to other agents' policies and allowing the gradient flow through communication channel. 3) TarMAC (Das et al. (2019)): we adopted the key concept of TarMAC in which the agent sends targeted messages using a signature-based attention model. 4) Comm-OA: the message consists of its own observation and action. 5) ATOC (Jiang & Lu (2018)): an attentional communication model which learns when communication is needed and how to combine the information of agents. We considered three multi-agent environments: predator-prey, cooperative navigation, and traffic junction, and we slightly modified the conventional environments to require more coordination among agents.

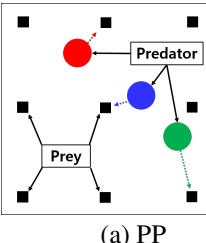 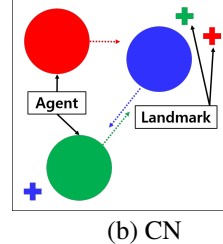 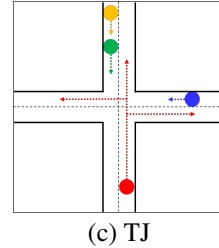

(a) PP      (b) CN      (c) TJ

Figure 2: Considered environments: (a) predator-prey (PP), (b) cooperative-navigation (CN), and (c) traffic-junction (TJ)

## 5.1 ENVIRONMENTS

**Predator-prey (PP)** The predator-prey environment is a standard task in multi-agent systems. We used a PP environment that consists of $N$ predators and fixed $M$ preys in a continuous state-action domain. We control the actions of predators and the goal is to capture as many preys as possible in a given time. Each agent observes the positions of predators and preys. When $C$ predators catch a prey simultaneously, the prey is captured and all predators get shared reward $R_1$. At every time when all the preys are captured, the preys are respawn and the shared reward value $R_1$ increases by one with initial value one to accelerate the capture speed for the given time. We simulated three cases: $(N = 2, C = 1)$, $(N = 3, C = 1)$, and $(N = 4, C = 2)$ with all $M = 9$ preys, where the fixed positions of the preys are shown in Fig.2(a). In the cases of $(N = 2, C = 1)$ and $(N = 3, C = 1)$, the initial positions of all predators are the same and randomly determined. Thus, the predators should learn not only how to capture preys but also how to spread out. In the case of $(N = 4, C = 2)$, the initial positions of all predators are randomly determined independently. Thus, the predators should learn to capture preys in group of two.

**Cooperative-navigation (CN)** The goal of cooperative navigation introduced in (Lowe et al. (2017)) is for $N$ agents to cover $L$ landmarks while avoiding collisions among the agents. We modified the original environment so that collisions occur more easily. We set $L = N$, increased the size of agent, and assigned a specific landmark to cover to each agent (i.e., each agent should cover the landmark of the same color in Fig.2(b)). Each agent observes the positions of predators and landmarks. The agent receives shared reward $R_1$ which is the sum of the distance between each agent and the corresponding landmark at each time step and success reward $N' \times R_2$ where $N'$ is the number of the covered landmark. Agents who collide with other agents receive negative reward $R_3$. We simulated the environment with $N = L = 3$, $R_1 = 1/3$, $R_2 = 1$ and $R_3 = -5$.

**Traffic-junction (TJ)** We modified the traffic-junction introduced in Sukhbaatar et al. (2016) to continuous state-action domain. In the beginning of an episode, each agent is randomly located in a predefined initial position and assigned one of three routes: left, right or straight, as seen in Fig.2(c). The observation of each agent consists of the positions of all agents (no route information of other agents) and 2 one-hot vectors which encodes the initial position and assigned route of the agent. The action of each agent is a real value in $(0, 1)$, which indicates the distance to go along the assigned route from the current position. The goal is to go to the destination as fast as possible while avoiding collision with other agents. To achieve the goal, we design reward with three components. Each agent receives success reward $R_1$ if it arrives at the destination without any collision with

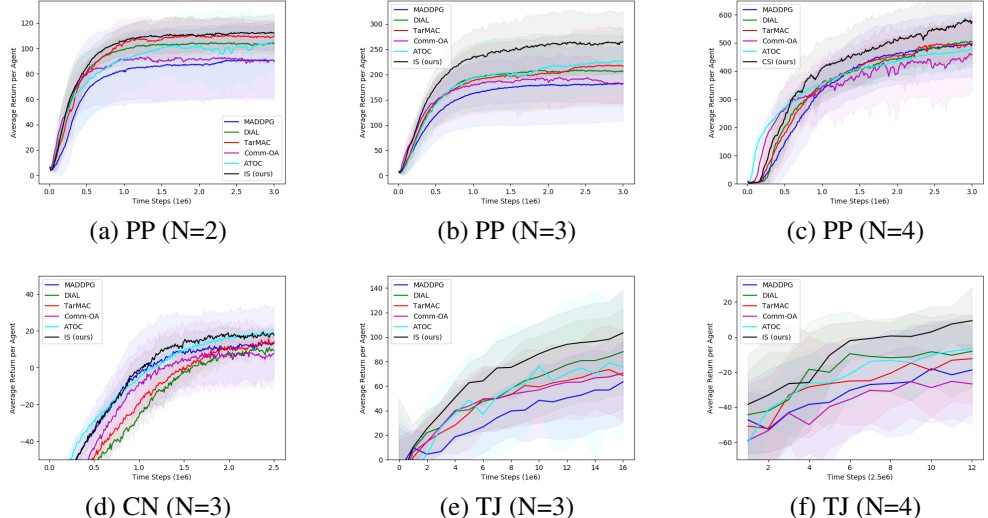

Figure 3: Performance for MADDPG (blue), DIAL (green), TarMAC (red), Comm-OA (purple), ATOC (cyan) and the proposed IS method (black).

other agents, collision negative reward $R_2$ if its position is overlapped with that of other agent, and time negative reward $R_3$ to avoid traffic jam. When an agent arrives at the destination, the agent is assigned a new initial position and the route. An episode ends when $T$ time steps elapse. We set $R_1 = 20$, $R_2 = -10$, and $R_3 = -0.01\tau$, where $\tau$ is the total time step after agent is initialized.

## 5.2 RESULTS

Fig. 3 shows the performance of the proposed IS scheme and the considered baselines on the PP, CN, and TJ environments. Figs.3(a)-(d) shows the learning curves of the algorithms on PP and CN and Figs. (e)-(f) show the average return using deterministic policy over 100 episodes every 250000 time steps. All performance is averaged over 10 different seeds. It is seen that Comm-OA performs similarly to MADDPG in the considered environments. Since the received messages come from other agents at the previous time step, Comm-OA in which the communication message consists of agent's observation and action performs similarly to MADDPG. Unlike Comm-OA, DIAL, TarMAC, and ATOC outperform MADDPG and the performance gain comes from the benefit of learning communication protocol in the considered environments except PP with $N = 4$. In PP with $N = 4$, four agents need to coordinate to spread out in group of two to capture preys. In this complicated coordination requirement, simply learning communication protocol based on past and current information did not obtain benefit from communication. On the contrary, the proposed IS scheme sharing intention with other agents achieved the required coordination even in this complicated environment.

## 5.3 ANALYSIS

**Imagined trajectory** The proposed IS scheme uses the encoded imagined trajectory as the message content. Each agent rolls out an imagined trajectory based on its own policy and trained models including action predictor and observation predictor. Since the access to other agents' policies is not available, the true trajectory and the imagined trajectory can mismatch. Especially, the mismatch is large in the beginning of an episode because each agent does not receive any messages from other agents (In this case, we inject zero vector instead of the received messages into the policy). We expect that the mismatch will gradually decrease as the episode progresses and this can be interpreted as the procedure of coordination among agents. Fig.5 shows the positions of all agents and each agent's imagined trajectory over time step in one episode for predator-prey with $N = 3$ predators, where the initial positions of the agents after the end of training ($t = 0$) is bottom right on the map. Note that each agent estimates the future positions of other agents as well as their

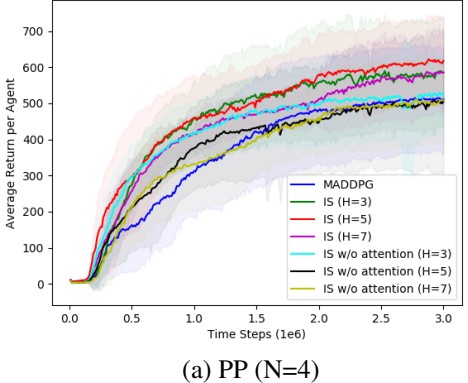
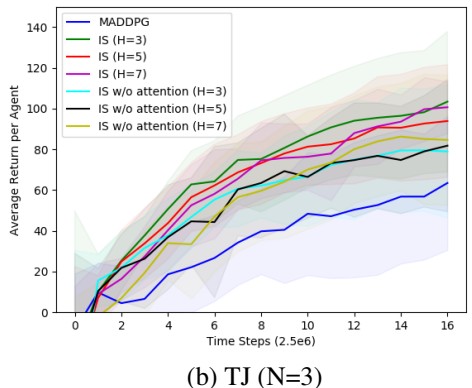

(a) PP (N=4)
(b) TJ (N=3)

Figure 4: Performance for our proposed method with different length of imagined trajectory $H$ and without attention module.

own future position due to the assumption of full observability. The first, second, and third row of Fig.5 show the imagined trajectories of all agents at Agent 1 (red), Agent 2 (green) and Agent 3 (blue), respectively. Note that the imagined trajectory of each agent represents its future plan for the environment. As seen in Fig.5, at $t = 0$ the intention of both Agent 1 and Agent 3 is to move to the left to catch preys. At $t = 1$, all agents receive the messages from other agents. It is observed that Agent 3 changes its future plan to catch preys around the center while Agent 1 maintains its future plan. This procedure shows that coordination between Agent 1 and Agent 3 starts to occur. It is seen that as time goes, each agent roughly predicts other agents' future actions.

We conducted experiments to examine the impact of the length of the imagined trajectory $H$. Fig.4 shows the performances of the proposed method for different values of $H$. It is seen that the training speed is reduced when $H = 7$ as compared to $H = 3$ or $H = 5$. However, the final performance all outperforms the baseline.

**Attention** In the proposed IS scheme, the imagined trajectory is encoded based on the attention module to capture the importance of components in the imagined trajectory. Recall that the message of Agent $i$ is expressed as $m_t^i = \sum_{h=1}^{H} \alpha_h^i v_t^{i,h}$, as seen in (7), where $\alpha_h^i$ denotes the importance of $v_t^{i,h}$, which is the encoded imagined step. Note that the previously proposed communication schemes are the special case corresponding to $\alpha^i = (1, 0, \cdots, 0)$. In Fig.5, the brightness of each circle is proportional to the attention weight. At time step $t = K$, where $K = 37$, $\alpha_2^1$, which indicates when Agent 1 moves to the prey in the bottom middle, is the highest. In addition, $\alpha_4^3$, which indicates when Agent 3 moves to the prey in the left middle, is the highest. Hence, the agent tends to send future information when it is near a prey. Similar attention weight tendency is also captured in the time step $t = K + 1$ and $t = K + 2$.

As aforementioned, the aim of the IS scheme is to communicate with other agents based on their own future plans. How far future is important depends on the environment and on the tasks. In order to analyze the tendency in the importance of future plans, we averaged the attention weight over the trajectories on the fully observable PP environment with 3 agents and a partially observable PP environment with 3 agents in which each agent knows the locations of other agents within a certain range from the agent. The result is summarized in Table 1. It is observed that the current information (time $k$) and the farthest future information (time $k + 4$) are mainly used as the message content in the fully observable case, whereas the current information and the information next to the present (time $k$ and $k + 1$) are mainly used in the partially observable environment. This is because sharing observation information is more critical in the partially observable case than the fully observable case. A key aspect of the proposed IS scheme is that it adaptively selects most important steps as the message content depending on the environment by using the attention module.

We conducted an ablation study for the attention module and the result is shown in Fig. 4. We compared the proposed IS scheme with and without the attention module. We replace the attention

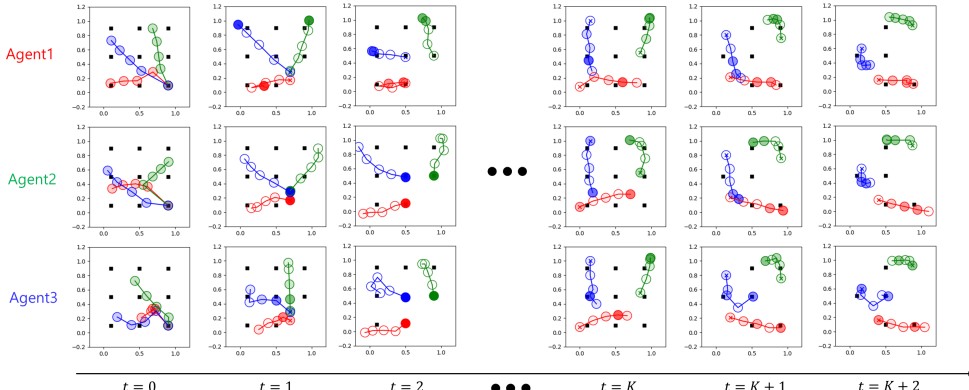

Figure 5: Imagined trajectories and attention weights of each agent on PP (N=3): 1st row - agent1 (red), 2nd row - agent2 (green), and 3rd row - agent3 (blue). Black squares, circle inside the times-icon, and other circles denote the prey, current position, and estimated future positions, respectively. The brightness of the circle is proportional to the attention weight. $K = 37$.

module with an averaging layer, which is the special case corresponding to $\alpha^i = (\frac{1}{H}, \cdots, \frac{1}{H})$. Fig. 4 shows that the proposed IS scheme with the attention module yields better performance than the one without the attention module. This shows the necessity of the attention module. In the PP environment with 4 agents, the imagined trajectory alone without the attention module improves the training speed while the final performance is similar to that of MADDPG. In the TJ environment with 3 agents, the imagined trajectory alone without the attention module improves both the final performance and the training speed.

| Imagined steps | k | k+1 | k+2 | k+3 | k+4 |
|---|---|---|---|---|---|
| Fully observable PP ($N$=3) | **0.33** | 0.18 | 0.15 | 0.14 | **0.20** |
| Partially observable PP ($N$=3) | **0.32** | **0.22** | 0.17 | 0.15 | 0.14 |

Table 1: Averaged attention weight over the trajectory at time step $k$

## 6 CONCLUSION

In this paper, we proposed the IS scheme, a new communication protocol, based on sharing intention among multiple agents for MARL. The message-generation network in the proposed IS scheme consists of ITGM, which is used for producing predicted future trajectories, and AM, which learns the importance of imagined steps based on the received messages. The message in the proposed scheme is encoded imagined trajectory capturing the agent's intention so that the communication message includes the future information as well as the current information, and their weights are adaptively determined depending on the environment. We studied examples of imagined trajectories and attention weights. It is observed that the proposed IS scheme generates meaningful imagined trajectories and attention weights. Numerical results show that the proposed IS scheme outperforms other communication algorithms including state-of-the-art algorithms. Furthermore, we expect that the key idea of the proposed IS scheme combining with other communication algorithms such as ATOC and TarMAC would yield even better performance.

## 7 ACKNOWLEDGMENTS

This research was supported by Basic Science Research Program through the National Research Foundation of Korea (NRF) funded by the Ministry of Science, ICT & Future Planning(NRF-2017R1E1A1A03070788).

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

## A MULTI-AGENT DEEP DETERMINISTIC POLICY GRADIENT (MADDPG)

MADDPG is an extended version of DDPG to multi-agent systems under the framework of CTDE (Lowe et al. (2017)). Each agent has a deterministic policy $a_t = \mu^i_{\theta_\mu}(o_t)$ conditioned on its own observation $o_t$ and a centralized critic $Q^i_{\theta_Q}(x, a) = E[R^i_t | x_t = x, a_t = a]$ conditioned on the joint actions $a_t$ and state information $x_t$. Here, $x_t$ can be state $s_t$ or the set of observations $(o^1_t, \cdots, o^N_t)$. The centralized critic is trained by minimizing the following loss:

$$L_Q(\theta_Q) = \mathbb{E}_{x,a,r^i,x'}\Big[(y^i - Q^i_{\theta_Q}(x, a))^2\Big], \quad y^i = r^i + \gamma Q^i_{\theta_Q^-}(x', a')|_{a^{j'}=\mu^{i-}(o^j)}, \quad (14)$$

where $\theta_Q^-$ is the parameter of target Q-function and $\mu^{i-}$ is the target policy of Agent $i$. The policy is trained by Deterministic Policy Gradient (DPG), and the gradient of the objective with respect to the policy parameter $\theta_{\mu^i}$ is given by

$$\nabla_{\theta_{\mu^i}} J(\mu^i) = \mathbb{E}_{x,a}\Big[\nabla_{\theta_{\mu^i}} \mu^i(o^i) \nabla_{a^i} Q^i_{\theta_Q}(x, a)|_{a^i=\mu^i(o^i)}\Big]. \quad (15)$$

# B  TRAINING DETAILS AND HYPERPARAMETERS

Table 2: Hyperparameters of all algorithms

|  | MADDPG | TARMAC | DIAL | ATOC | IS |
|---|---|---|---|---|---|
| REPLAY BUFFER SIZE | $2 \times 10^5$ | $2 \times 10^5$ | $2 \times 10^5$ | $2 \times 10^5$ | $2 \times 10^5$ |
| DISCOUNT FACTOR | 0.99 | 0.99 | 0.99 | 0.99 | 0.99 |
| MINI-BATCH SIZE | 128 | 128 | 128 | 128 | 128 |
| OPTIMIZER | ADAM | ADAM | ADAM | ADAM | ADAM |
| LEARNING RATE | 0.0005 | 0.0005 | 0.0005 | 0.0005 | 0.0005 |
| NUMBER OF HIDDEN LAYERS (ALL NETWORKS) | 2 | 2 | 2 | 2 | 2 |
| NUMBER OF HIDDEN UNITS PER LAYER | 128 | 128 | 128 | 128 | 128 |
| ACTIVATION FUNCTION FOR HIDDEN LAYER | RELU | RELU | RELU | RELU | RELU |
| MESSAGE DIMENSION ON PP | - | 5 | 5 | 5 | 5 |
| MESSAGE DIMENSION ON CN | - | 3 | 3 | 3 | 3 |
| MESSAGE DIMENSION ON TJ | - | 3 | 3 | 3 | 3 |
| ATTENTION DIMENSION ON PP | - | 5 | - | 5 | 5 |
| ATTENTION DIMENSION ON CJ | - | 3 | - | 3 | 3 |
| ATTENTION DIMENSION ON TJ | - | 3 | - | 3 | 3 |
| IMAGINED TRAJECTORY LENGTH H | - | - | - | - | 5 |

## C  ADDITIONAL ABLATION STUDY

We conducted an additional experiment to examine whether performance improvement is gained from sharing intention or having a prediction of the future. We compared the proposed IS scheme with MADPPG-p in which the agent does not use communication, but uses their own imagined trajectory as additional input. Fig. 6 shows that the proposed IS scheme outperforms MADDPG-p. Thus, sharing intention, which is a core idea of this paper, is more important than having a prediction of the future.

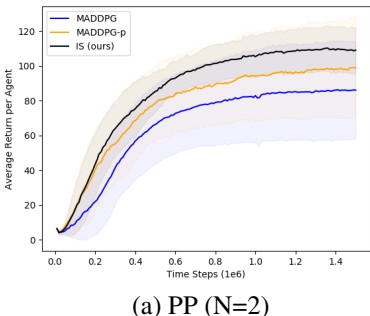
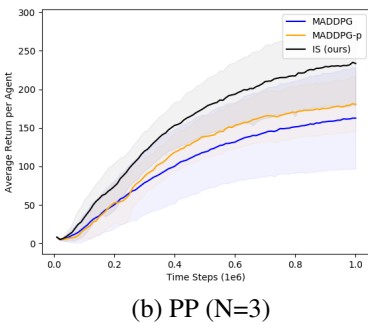

(a) PP (N=2)  (b) PP (N=3)

Figure 6: Performance for MADDPG (blue), MADDPG-p (orange), and the proposed IS method (black).

# D    PSEUDO CODE

---

**Algorithm 1** Intention Sharing (IS) Communication Scheme

---

Initialize parameter $\theta_\mu^i, \theta_Q^i, \theta_\mu^{i-}, \theta_Q^{i-}, \theta_o^i, \theta_a^i, W^i, \; \forall i \in \{1, \cdots, N\}$
**for** $episode = 1, 2, \cdots$ **do**
    Initialize state $s_1$, messages $m_0 = (\overrightarrow{0}, \cdots, \overrightarrow{0})$ and each agent observes $o_1^i$
    **for** $t <= T$ and $s_t \neq$ terminal **do**
        Each agent receives the messages $m_{t-1} = (m_{t-1}^1, \cdots, m_{t-1}^N)$
        Each agent selects action $a_t^i \sim \pi^i(\cdot | o_t^i, m_{t-1})$ for each agent $i$
        Execute $\boldsymbol{a_t}$ and each agent $i$ receives $r_t$ and $o_{t+1}^i$
        **for** $h = 1, 2, \cdots, H$ **do**
            Predict other agents' actions $\hat{a}_{t+h-1}^{-i}$ from the action predictor $f_a^i$
            Generate $\hat{o}_{t+h}^i$ from observation predictor $f_o^i(o_{t+h-1}^i, \hat{a}_{t+h-1}^i, \hat{a}_{t+h-1}^{-i})$
            Generate $\hat{a}_{t+h}^i \sim \pi^i(\cdot | \hat{o}_{t+h}^i, m_{t-1})$
        **end for**
        Each agent generates the messages $m_t^i$ by injecting $\tau^i = (\tau_t^i, \hat{\tau}_{t+1}^i, \cdots, \hat{\tau}_{t+H-1}^i)$ into Attention Module (AM)
        Store transitions in $D$
    **end for**
    **for** each gradient step **do**
        Update $\theta_Q^i$ and $(\theta_o^i, \theta_a^i)$ by minimizing the loss (9) and the loss (12)
        Update $\theta_\mu^i$ and $W^i$ based on the gradient (10) and the gradient (11)
    **end for**
    Update $\theta_\mu^{i-}, \theta_Q^{i-}$ using the moving average method
**end for**

---

