# OpenReview forum: "Communication in Multi-Agent Reinforcement Learning: Intention Sharing"
_ICLR.cc/2021/Conference — ICLR 2021 Poster_

### Official Review · AnonReviewer3 · 2020-10-27
**Interesting method for an important problem. Would be improved by ablations to know what matters.**

**Rating:** 6
**Confidence:** 4

**Review:**

---- Summary ----
The paper proposes a new method for training communication in multiagent systems. The method involves agents producing imagined future trajectories using learned environment dynamics, and then communicating some parts of these trajectories to other agents. This communication scheme is evaluated on several cooperative communication domains, outperforming other methods.

---- Reasons for score ----
The paper addresses an important problem, the method is sensible, and the empirical results suggest the method proposed works. My main concern is a lack of ablations, which makes it hard to tell which parts of the method are important.

---- Pros ----
1. The problem is well motivated. The existing literature on learned multi-agent communication primarily addresses the problem of communicating information about partially observed states. However, this is not all there is to communication, and the communication of agent intentions is an important topic to address.

2. The method suggested - basing messages on predicted future trajectories - is sensible and well motivated.

3. The experiments compare the method to a good range of baseline algorithms for multiagent communication, and it performs well.

---- Cons/Questions ----
The method proposed has several moving parts, and without ablations it is hard to tell which are the most important pieces. There are a few experiments I would like to see added to work out what is important in the method:
* A comparison to the uniform attention case. While the qualitative analysis suggests the attention mechanism is meaningfully used, it would be good to show how this affects performance.
* An ablation where all the attention is put on the first timestep, to show the effect of the imagined trajectories. While this is similar to the other communication methods discussed in the paper, I am not sure whether it directly corresponds to any of them.
* A comparison to a case where the agents receive only their own messages to condition their next actions. At the moment is unclear whether the advantage is gained from sharing intentions between agents, or by having a prediction of the future.

All the environments used are modifications of existing environments; was there a reason to change these? Particularly, the Cooperative-Navigation environment looks like it could have been used as-is, for an easier comparison to previous work?

---- Post-rebuttal ----
The additional ablation studies on the attention and MADDPG-p are helpful, and address most of my concerns - though it is still not clear to me whether any of the baselines compared to is exactly equivalent to the method used with attention weights (1, 0, 0, ...). I share the concerns of Reviewer 4 about the significance of the results - this is still not in the paper, and not present for the new experiments. Overall, I have not changed my rating.

---

> ### Author Response · Authors · 2020-11-23
> **Response to Reviewer 4**
>
> Thank you for the valuable comments. Here is our reply.
>
> Regarding additional ablation studies:
> - As mentioned in the common response, we added ablation studies regarding AM and the length of the imagined trajectory H. Please see Fig. 4 in the revised paper.
>
>
> Regarding the modifications of existing environments:
> - We modified the existing environments to  require more coordination among agents. As mentioned in the main paper, the agents in the PP environment should learn to spread out as well as capture the preys in order to gain more rewards. In addition, we increased the size of the agent in the CN environment so that the agent has to be more careful not to crash with other agents.
>
> Regarding "all the attention is put on the first timestep":
> - As mentioned in the main paper, the previously proposed communication schemes are the special case corresponding to $\alpha=(1,0,\cdots, 0)$, which means that all the attention is put on the first timestep. Especially, DIAL sends the learned features of the information in the first time step as messages.
>
> Regarding "A comparison to a case where the agents receive only their own messages to condition their next actions."
> - We compared the proposed IS scheme with the case in which the agents receive only their own messages to condition their next actions, and the result is provided in Appendix D. It is seen that MADDPG-p in which each agent uses own imagined trajectory as an additional input yields better performance as compared to the MADDPG baseline. However, the proposed IS scheme outperforms MADDPG-p. Thus, sharing intention, which is a core idea of this paper, is more important than having a prediction of the future.

---

### Official Review · AnonReviewer1 · 2020-10-27
**Review for paper "Communication in Multi-Agent Reinforcement Learning: Intention Sharing"**

**Rating:** 4
**Confidence:** 3

**Review:**

[Summary]

Paper proposed to generate the communication message in MARL with the predicted trajectories of all the agents (include the agent itself). An extra self-attention model is also stacked over the trajectories to trade off the length of prediction and the possible explaining away issue.  The whole model is trained via a canonical MARL objective while the trajectory prediction model utilizes direct supervision collected from the environments. Experiments on several toy MARL benchmark demonstrates the effectiveness of the proposed method.

[Strengh]

+) The idea of communication with imagined intention is motivated properly with rich psychological background and also technically sound.

+) The paper is overall clear and well-written. I found there are enough technical details to reproduce the main results of the main approach.

+) Still limited though in terms of the converted domains, the empirical evaluations deliver impressive results over a reasonable collection of counterparts.

[Weakness]

The main concerns I have with this submission lies in the overall novelty and the evaluation of the proposed method. After reading this paper, indeed I find the authors failed to capture some important research in this narrow area and it's still not clear how does the proposed method really works and whether it is sensitive to some specific implementing factors.

-) The idea of intention sharing based on prediction is essentially not novel esp. in the MARL domain. In the CogSci & AI community, theory-of-mind (ToM) and its application in multi-agent execution (either collaborative or zero-sum games) have been well studied for years [1, 2, 3]. Although I may agree that there are few prior works [4] on sending these predictions as messages to other agents, it does not really introduce that many new ideas to how collaborative or adversarial agents could benefit from such ToM-based intention prediction. However, the authors failed to capture these counterparts in their discussion and evaluations. Specifically, I would like to see how does the proposed method formulate its intention prediction differently than the prior work and whether it could enjoy advantages in performance with such differences.

-) The proposed method is essentially quite complex (stacked prediction, transformer, etc) than its Bayesian counterparts, while the authors only provide an overall evaluation against several MARL baselines. It will be critical to also conduct a serious ablation study given the overall complexity of the proposed method, i.e. whether to use the transformer, the architectural choice of the transformer (num. of heads, etc), and the length of predicted trajectories (H). Also, the disagreement between the predicted trajectories and actual observation and its relation to the performance should also be investigated.

-) Although the selected tasks are all canonical to MARL, given the fact a growing number of recent MARL learners have been moved on to more challenging tasks, I feel it would be necessary to include some mini MOBA games or other tasks with vision-based observations.

[1] Bayesian models of human action understanding

[2] Theory-based Social Goal Inference

[3] Bayesian Theory of Mind: Modeling Joint Belief-Desire Attribution

[4] Machine theory of mind

[Suggestions&Questions]

(1) Add ablation studies on the use of self-attention model and the length of prediction (H).

(2) Visualize&compare the predicted and observed trajectories, add some discussion on how such disagreement would affect the performances.

(3) (minor) Try the proposed method on mini MOBA games or tasks with vision-based observations.

(4) Add citations to the related work on ToM and its application in MAS.

[Post-rebuttal]

I have read through all the other reviews and the rebuttal. Would like to thank the authors for their efforts in improving this submission. However, the main issue on the lack of novelty remains and I also find R4's concern on the significance of results is valid. Therefore, I will keep my initial justification as is.

---

> ### Author Response · Authors · 2020-11-23
> **Response to Reviewer 3**
>
> Thank you for the valuable comments. Here is our reply:
>
> Regarding the novelty of our proposed method:
> - We agree on that the notion of intention was studied in the MARL community. For example, in Theory of Mind (ToM) and Opponent Modeling (OM), each agent  infers other agents' goal or action or belief, which captures  other agents' intention.
>     It has been shown in these works that the inference of other agents' goal or action or belief can improve the performance. In our work, instead of inference, we use communication to share the agents' intention. Unlike the above inference-based  approaches, the agent in our approach generates their own intention and use it as the messages. In addition, the previous approaches infer the current (or just next-time step) information of other agents.     However, we use the imagined trajectory, which captures the future behavior far beyond the current time step. To the best of our knowledge, our work is the first that introduces communication for sharing the intention among agents.
>
> Regarding ablation studies on AM and the length of the imagined trajectory H:
> - As mentioned in the common response, we added ablation studies regarding the attention module and the length of imagined trajectory H. Please see Fig. 4 in the revised paper.
>
> Regarding "Visualize and compare the predicted and observed trajectories, add some discussion on how such disagreement would affect the performances.":
> - We have already provided a visualization of the imagined trajectory and the observed trajectory in Fig. 5. It is difficult to precisely analyze the influence of the disagreement on the performance, but it is observed in Fig. 5 that the disagreement often occurs in the early stage of episode since the agents do not receive any messages from other agents at the first step of each episode.  It is also observed in Fig. 5 that the disagreement decreases as time goes. This means that the agents know other agents' plan

---

### Official Review · AnonReviewer2 · 2020-10-28
**well-written and technically sounds**

**Rating:** 6
**Confidence:** 3

**Review:**

Summary: This work proposes to use 'intention' of each agent to enhance the message sharing scheme of MARL. For training, MADDPG is used as a backbone MARL and implement ITGM and AM on top of it. The proposed method is compared with several baseline approaches from three different environments.

Strengths:
+ The paper is well-written and technically sounds.
+ The motivation is clear.

Weaknesses:
- This work wants to see the effectiveness of the use of intention as a communication scheme under MARL. However, in general, this is not the first to use intention. I suggest to review some relevant works and show how 'intention' has been explored in a similar/different way in the literature.
- Although visualization of imagined trajectory is given in Figure 4, it does not fully demonstrate the idea of this approach. More qualitative evaluation seems required to validate from various aspects.
- It seems that attention is used to capture which waypoint is more important in an imagined trajectory rather than whose imagined trajectory is more important. If so, is it really important? Where is the ablative study of using attention?
- Intuitively, when two agents set the same goal (catching same prey), what is the rationale of determining who maintain the goal or who change the plan?

---

> ### Author Response · Authors · 2020-11-23
> **Response to Reviewer 2**
>
> Thank you for the valuable comments. Here is our reply:
>
> Regarding the use of intention in the literature:
> - As mentioned in the common response, we added  Theory of Mind (ToM) and Opponent Modeling (OM), which use the notion of intention, in the related work section. However, please see the common response regarding the difference of our method from these existing methods.
>
> Regarding more ablation studies:
> - As mentioned in the common response, we added ablation studies regarding AM and the length of imagined trajectory H into the revised paper.
>
> Regarding the rationale of determining who maintain the goal or who change the plan:
> - Intuitively, we think that some agents whose policy is sensitive to the received messages (e.g. the weights connected to the received messages are quite large) will be trained to be the agents who tend to change the plan, other agents whose policy is less sensitive to the received messages will be trained to be the agents who tend to maintain the plan.

---

### Official Review · AnonReviewer4 · 2020-10-29
**Paper is well written, clearly explains the proposed approach, compares to appropriate state-of-the-art baseline methods, but there is significant room for improvement in the experimental results**

**Rating:** 5
**Confidence:** 4

**Review:**

Summarization and Strengths:
This paper studies how to learn coordinated behavior among multiple agents by learning a communication protocol. Specifically, compared to existing works, this paper proposes to generate messages based on not only current information but also future information (referred to as imagined trajectory) (Section 4.1). Additionally, the attention module in Section 4.2 is learned to weigh between current and future information dynamically. Overall, the paper is well written, clearly explains the proposed approach, and compares to appropriate state-of-the-art baseline methods.

Weaknesses:
While the motivation of this work is clear, the experimental results fail to support the claim. The results in Figure 3 show a small performance difference between IS (proposed approach) and baselines due to the large variance. Performing a statistical test, such as the t-test, to verify whether the proposed approach achieves statistically significant results than baselines will be helpful.

Further Questions:
1. In general, learning the attention mechanism improves results, but is it necessary for the proposed approach? Adding an abbreviation analysis, such as the performance difference between with and without attention, will be helpful.
2. The action predictor (Equation (2)) predicts the peer agents' actions based on the agent's own observation. While the peer action prediction based on the agent's own observation supports the decentralized execution, the local observation may not include sufficient information about the others depending on a multiagent domain and thus the action prediction can fail. Instead, wouldn't it be better to predict the peer agents' actions based on both the agent's own observation and received message m_{t-1} because the received message can contain useful information about the peers' behaviors?
3. As noted in footnote 1, the message at m_{t-1} is used for all H prediction steps. How does the performance differ with respect to different H values?

Minor comments:
1. Typo in Page 1: ''How to harness ... partial observation.'' -> ``How to harness ... partial observation.''
2. In Equation (2), the subscript "t" is missing (e.g., \hat{a}^{i-1} -> \hat{a}^{i-1}_{t})
3. Adding pseudo algorithm in Appendix will further clarify the proposed approach
4. For PP domain explanation (Section 5.1), it is unclear whether prey can run away from predators. In general, preys can run away in PP, but in this paper, preys seem to be fixed and cannot move around.
5. For PP and CN domain explanation (Section 5.1), it is unclear what each agent observes.
6. The intention prediction is also related to the theory of mind and opponent modeling (e.g., He et al., ICML-16, Raileanu et al., ICML-18, Rabinowitz et al., ICML-18), which can be added in the related work section (Section 2).

He He, Jordan Boyd-Graber, Kevin Kwok, Hal Daumé III. Opponent Modeling in Deep Reinforcement Learning. ICML-16

Roberta Raileanu, Emily Denton, Arthur Szlam, Rob Fergus. Modeling Others using Oneself in Multi-Agent Reinforcement Learning. ICML-18

Neil C. Rabinowitz, Frank Perbet, H. Francis Song, Chiyuan Zhang, S.M. Ali Eslami, Matthew Botvinick. Machine Theory of Mind. ICML-18

I have read over the rebuttal and discussion and will keep my evaluation score as it was since the concerns about the weak performance result still remain.

---

> ### Author Response · Authors · 2020-11-23
> **Response to Reviewer 1**
>
> Thank you for the valuable comments. As mentioned in the common response, we added ablation studies regarding AM and the length of the imagined trajectory. In addition, we have included ToM and OM in the revised related work section. Here is our reply:
>
> Regarding the action predictor:
> -  We agree on that the received messages can be helpful to predict other agents' actions, especially in partially observable environments. The action predictor in our approach takes only its own observation as input and is trained in the manner of supervised learning. However, the action predictor can be replaced with the previously proposed opponent modeling approach and can take the received messages, as the reviewer mentioned. We added the flexibility of choosing one of other methods as the action predictor into the revised paper.
>
> Regarding the PP and CN environment:
> - The preys in our PP environment are fixed. We focus on the predators: The predators should learn to spread out in order to capture the prey ($N=2, N=3$) and the predators in the PP ($N=4$) should learn to capture prey in a group of two. The agent in the PP environment observes the positions of predators and preys and the agent of the CN environment observes the positions of agents and landmarks.
>
> Regarding a statistical test:
> - We used deterministic evaluation based on 20 episodes generated by the corresponding deterministic policy after the training ends. We conducted a pairwise t-test to verify the proposed IS scheme on the PP environment: (a) In the PP (N=2) environment, the proposed IS scheme outperforms MADDPG and Comm-OA with 95\% confidence level (CL) and outperforms DIAL and ATOC with 80\% confidence level whereas the performance of TarMAC is similar with the proposed IS scheme. (b) In the PP (N=3) environment, the proposed IS scheme outperforms all baselines with 99\% confidence level. (c) In the PP (N=4) environment, the proposed IS scheme outperforms Comm-OA, ATOC, and TarMAC with 95\% confidence level and outperforms MADDPG and DIAL with 90\% confidence level.

---

### Author Response · Authors · 2020-11-23
**Common Response**

We thank all reviewers for the valuable comments. We revised our paper based on the reviewers' comments and here is our response to the common comments.

- We added ablation studies on the Attention Module (AM) and the length of the imagined trajectory H as Fig. 4.  It is observed that ITGM without AM increases the training speed as compared to the MADDPG baseline. The final performance by ITGM without AM is almost the same as that of the baseline in the PP environment whereas the final performance by ITGM without AM increases in the TJ environment. Please see Fig. 4 in the revised paper. Furthremore, if we add AM to ITGM, then both the training speed and the final performance are increased. Hence, using both ITGM and AM is important.

- We added Theory of Mind (ToM) and Opponent Modeling (OM) in the related work section. We expect that ToM and OM can be used for our action predictor. We summarized the difference between the proposed IS scheme and (ToM, OM) as follows:
    (a) Our approach uses communication to share the intention, whereas ToM and OM infer other agents' goal or action capturing the intention.
     (b) Our approach uses future information by rolling-out the policy, whereas ToM and OM infer only the current or just next time-step information.
- We conducted an ablation study to examine whether performance improvement is gained from sharing intention or having a prediction of the future in Appendix C.
-  We added the pseudo-code in Appendix D.
-  We conducted the PP environment with additional 10 different seeds (The performance is averaged over 20 different seeds).

---

### Decision · Program_Chairs · 2021-01-07
**Final Decision**

**Decision:**

Accept (Poster)

**Comment:**

The authors study co-ordination in multi-agent systems. Specifically they propose a scheme where agents model future trajectories through the environment dynamics and other agents' actions, they then use this to form a plan which forms the agents' intention which is then communicated to the other agents.

The major concerns raised by the reviewers were around novelty, lack of ablations and significance of results as improvements were modest. During the rebuttal, the authors have extended their work with ablations and have conducted a statistical test. While it is true the current results present a small improvement, i think this is an interesting contribution in the field of emergent communication

---

> ### Comment · ~Tiong_Bernard1 · 2023-06-04
> **Relationship between the proposed and emergent communication**
>
> May I know how the proposed method is related to emergent communication?